# One-Stage Hip Revision Arthroplasty Using Megaprosthesis in Severe Bone Loss of The Proximal Femur Due to Radiological Diffuse Osteomyelitis

**DOI:** 10.3390/tropicalmed7010005

**Published:** 2021-12-31

**Authors:** Roy Gonzalez, Ernesto Muñoz-Mahamud, Guillem Bori

**Affiliations:** 1Department of Traumatology and Orthopedic Surgery, Hospital San Juan de Dios, San José 10102, Costa Rica; ragonzalz@ccss.sa.cr; 2Department of Traumatology and Orthopaedic Surgery, Hospital Clínic de Barcelona, University of Barcelona, Carrer Villarroel, 170, 08036 Barcelona, Spain; emunoz@clinic.cat; 3Insitut d’Investigacions Biomèdiques August Pi i Sunyer (IDIBAPS), Carrer Rosselló, 149, 08036 Barcelona, Spain; 4Traumatology and Orthopaedic Surgery, University of Vic-Central University of Catalonia (UVic-UCC), 08500 Vic, Spain

**Keywords:** osteomyelitis, one-stage exchange arthroplasty, infection, hip, hip arthroplasty, bone defect, periprosthetic joint infection

## Abstract

Managing substantial proximal and/or distal femoral bone defects is one of the biggest challenges in chronic hip periprosthetic joint infection. Most authors use two-stage arthroplasty with a temporary antibiotic-loaded cement spacer for the management of these patients. In this study, we show our experience with one-stage exchange arthroplasty in managing severe bone defects due to radiological-extensive proximal femoral osteomyelitis. Two patients were included in the study. They showed radiological-extensive proximal femoral osteomyelitis, and they were treated with one-stage exchange arthroplasty using megaprosthesis. Diffuse osteomyelitis was confirmed in both cases; in one case, the histology was compatible with osteomyelitis, and the other case had a positive culture identified in a bone sample. At a minimum of a four-year follow-up, the patients did not reveal any clinical, radiological or laboratory signs of infection. In conclusion, one-stage exchange arthroplasty and megaprosthesis is an option for the treatment of chronic hip periprosthetic joint infection associated with radiological-diffuse proximal femoral osteomyelitis.

## 1. Introduction

Managing periprosthetic joint infection (PJI) has been a challenge for decades with high numbers of complications and failures reported in more than 20% of patients with a strict definition of treatment success [1]. As patients are living longer, treating PJI where there is already extensive bone loss due to stress shielding, osteolysis, or implant loosening [2], combined with defects produced during implant extraction and aggressive debridement during revision PJI arthroplasties, amplifies these challenges.

Currently, the standard management of chronic hip and knee prosthetic joint infection includes one-stage or two-stage revision, depending on different conditions (host and extremity status, surgeon experience, and pathogen). To date, there is no international gold standard treatment for chronic PJI of the hip; however, two-stage revision is the most widely used, and Spain is not an exception [3]. The literature review shows comparable results for one- and two-stage strategies [4,5], although this is not founded on comparative studies. It is clear that one-stage exchange arthroplasty has great advantages: the need for only one operative procedure, reduced hospitalization time, reduced overall costs, and relatively improved patient satisfaction, as long as treatment success is comparable. Two-stage arthroplasty with antibiotic-loaded cement has raised some concerns: it could hinder infection eradication by bacterial colonization of the spacer, and it seems that the local antimicrobial effect is negligible [6,7] in severe bone defect management; the use of bulky antibiotic-loaded cement could also produce instability with spacer migration, breakage, dislocation, and acetabular and femoral lysis and even fracture [8].

One-stage exchange revision arthroplasty for chronic PJI was described in Sweden and Germany 40 years ago [9,10]. There is sufficient evidence that supports using a one-stage exchange to eradicate PJI. An international consensus agreed upon the inclusion criteria for the one-stage exchange [11]; however, bone defects have been considered a one-stage exchange hip arthroplasty contraindication [12]. There is sparse reported experience on one-stage exchange hip arthroplasty for the management of chronic PJI cases with severe bone defects; conversely, two-stage exchange arthroplasty with different kinds of spacers as treatment in this particular scenario has been published.

Managing substantial proximal and/or distal femoral bone defects is one of the biggest challenges in chronic PJI. This implies two issues: first, proximal femoral bone absence entails distal prosthetic fixation on the femoral isthmus to achieve adequate fixation, and fixation would be even more compromised if the bone defect extends over the femoral diaphysis. Then, proximal femoral bone absence produces abductor musculature dysfunction and, therefore, worse clinical outcomes.

The aim of the present study is to show our experience with one-stage exchange arthroplasty in managing severe bone defects due to radiological extensive proximal femoral osteomyelitis.

## 2. Cases

### 2.1. Case 1

Patient One is an 80-year-old male with a history of cardiopathy. He received a left total hip replacement due to primary osteoarthritis, then suffered a Vancouver B2 periprosthetic femoral fracture in the third year from his primary procedure and required a revision of his total hip replacement. A year from the revision arthroplasty, he was transferred to our center due to chronic PJI. The patient presented with a productive fistula, C-reactive protein (CRP) of 20.4 mg/L, and an erythrocyte sedimentation rate (ESR) of 22 mm/h. The initial radiograph showed proximal femoral diffuse osteomyelitis and signs of non-union of his previous periprosthetic fracture (Figure 1). CT-guided periprosthetic samples and fistula draining material were taken and cultured; *Pseudomona areuginosa* grew in both. We decided on one-stage exchange revision arthroplasty with megaprosthesis: GMRS^®^ proximal femoral (Stryker, Kalamazoo, MI, USA) and trident^®^ acetabular shell with constrained liner (Stryker, Kalamazoo, MI, USA). Ceftazidime and Ciprofloxacin were administered 12 days before the revision arthroplasty to reduce the bacterial load. One of the six intraoperative cultures was positive for *P. areuginosa,* and histological analysis showed findings compatible with PJI (greater than five polymorphonuclear leukocytes (PMN) per high-power field in five high-power fields observed at 400× magnification). The patient was kept hospitalized for 19 days while receiving intravenous Ceftazidime and Ciprofloxacin, then received a full six weeks of oral Ciprofloxacin. At the six-year follow-up, the patient did not show any clinical, laboratory, or radiological signs of infection (Figure 2).

### 2.2. Case 2

Patient Two is a 52-year-old female with a history of chronic obstructive pulmonary disease; due to final stage osteoarthritis, a primary total hip replacement arthroplasty was implanted in her left hip. The patient pointed out that multiple hip prosthesis exchanges had been performed before arriving to our center; the last revision was carried out five years ago. On initial presentation, the patient had a productive fistula on the posterior thigh and a CRP of 63.2 mg/L. Diffuse osteomyelitis of the proximal femur and femoral stem rupture was evident on the initial radiograph (Figure 3). Cultures from CT-guided periprosthetic samples and fistula draining material were negative; therefore, she was not given antibiotic treatment before surgery. We carried out a one-stage revision arthroplasty with megaprosthesis: Megasystem-C stem Link^®^ (Hamburg, Germany) and G7^®^ acetabular system with freedom constrained liner (Zimmer Biomet, Delawar, USA). Four of seven intraoperative periprosthetic soft tissue samples were positive for *Staphylococcus epidermidis.* Femoral and acetabular bone samples were also cultured; both were positive for *S. epidermidis.* The histological analysis reported less than five polymorphonuclear leukocytes (PMN) per high-power field in five high-power fields observed at 400× magnification. The patient received extended empiric antibiotic coverage with Meropenem and Linezolid, which was started 15 min before surgery. The antibiotic coverage was changed to Linezolid upon identification of the microorganism, and she completed six weeks of treatment. The patient was hospitalized for 12 days. At the four-year follow-up, the patient did not reveal any clinical or laboratory signs of infection; radiography did not exhibit loosening or infection signs (Figure 4).

## 3. Discussion

In the previously reported cases of proximal femoral massive bone defects and associated osteomyelitis, we were able to successfully eradicate infection in one stage. Implant removal, massive resection of the compromised bone, extensive debridement of devitalized tissue, and antibiotic treatment were the basis for achieving the cure of infection. If we decided upon a two-stage treatment with an antibiotic-laden cement spacer, we could have risked an error by overconfidence in the local effect of antibiotic-loaded cement, by deciding to preserve some compromised bone to avoid a massive bone defect and thus failing to achieve eradication of infection. Therefore, we opted for extensive debridement of all compromised bone and tissue and megaprosthesis implantation in a one-stage treatment. Diffuse osteomyelitis was confirmed in both cases: in Case 1, the histology was compatible with osteomyelitis, and Case 2 had a positive culture identified in a bone sample.

Most authors use temporary antibiotic-loaded cement spacers for the management of chronic PJI in two-stage procedures; nevertheless, there are some authors who described two-stage revision hip arthroplasty without a temporary spacer implanted in the first stage [3]. Hipfl et al. [3] reported on 135 hip PJI cases: 28 patients had Paprosky femoral bone loss type ≥3A (14 patients, Paprosky type 3; 11 patients, Paprosky type 3B; and 3 patients, Paprosky type 4). The reported reinfection rate was 25% (7/28) in Paprosky femoral bone loss type ≥3A patients and 8.4% (9/107) in patients with femoral bone defect <3A according to the Paprosky classification. The reason for the difference in the reinfection rates shown was insufficient debridement of the osteolytic proximal femur and retention of devitalized bone with biofilm residues [3]. The authors avoided antibiotic-laden cement spacer implantation in complex and severe proximal femur bone defect cases due to the high mechanical complication rates presented, such as dislocation, periprosthetic fractures, and spacer fractures.

There is scant experience reported in the literature on megaprosthesis use in septic revision arthroplasty due to chronic PJI. Indications for revision arthroplasty using megaprosthesis for proximal femoral bone defect differ in the literature [1,13,14]; most reports show the results of megaprosthesis implantation due to aseptic loosening, periprosthetic fracture, and PJI, so it is hard to know exactly the megaprosthesis effectiveness in chronic PJI management, for either one- or two-stage revision arthroplasty.

In spite of this, there are some reports through which we could analyze the results of megaprosthesis use for massive proximal femoral bone defects in one-stage hip revision arthroplasty [1,15,16,17,18]. Corona et al. [15], in a cohort of 29 chronic PJI patients, described two one-stage septic revision hip arthroplasty cases: both had complications, one with a total femur arthroplasty sustained a hip dislocation, the other patient with recurrence of infection. Artico et al. [16] reported on five septic revision hip arthroplasty cases: one with one-stage revision suffered reinfection; the rest were managed with two-stage revision arthroplasty, and infection eradication was achieved in all of them. Ramappa et al. [17] reported on six PJI cases treated with one-stage revision; the hip was involved only in two of them, and total femur arthroplasty was used in both; there were no complications or recurrence of infection at the last follow-up (18 and 24 months), and patients reported no or mild pain. Grammatopoulos et al.’s [1] study is intriguing; they reported on 40 PJI patients managed with endoprosthetic replacement. Most (24/40) of the patients underwent two-stage revision arthroplasty; in the remaining 16, the prosthesis was implanted as a one-stage procedure. Reinfection or recurrence of infection was reported in seven patients (17%); in all of them, polymicrobial infection was identified. There was no difference in the rate of eradication of infection, re-operation, and functional outcome between the performed one- and two-stage hip revision arthroplasty. The Endoklinik group [18] recently published a study on resection of the proximal femur during one-stage review for infected hip arthroplasty. It is a case-control study, 57 patients who underwent one-stage revision arthroplasty for PJI of the hip and required resection of the proximal femur and the control group consisted of 57 patients undergoing one-stage revision without bony resection. They conclude that radical resection is associated with higher surgical complications and increased re-revision rates, as patients who required resection of the proximal femur were found to have a higher all-cause re-revision rate (29.8% versus 10.5%), largely due to reinfection (15.8% versus 0%). However, radical resection may be necessary and required for infection eradication, given the complexity of such septic revision arthroplasty. In these patients it is very important to avoid dislocation of the prosthesis, as in both our cases a constrained liner was implanted, as in this study they concluded that postoperative dislocation was associated with increased risk of subsequent re-revisions with an OR of 281.4. For this reason, Abdelaziz et al. [18] found that the use of dual mobility components/constrained liner in the resection group was higher than that of controls (94.7% versus 36.8%). An interesting fact of this study is how they explain that the decision to perform the resection of the proximal femur can be made preoperatively or intraoperatively. In both cases, it was decided preoperatively that one-stage revision arthroplasty would be performed with resection of the proximal femur due to radiographic imaging of osteomyelitis. The level of resection was decided intraoperatively based on macroscopic bone findings. We did not use any other system to decide the level of resection, neither the intraoperative bone frozen section [19,20] nor the fluorescent tetracycline bone labeling [21].

From a surgical point of view, we have used conventional megaprosthesis, without any special coating in order to prevent infection, despite the risk that these patients had. There are currently different antibacterial coatings of implants on the market that could be useful in those cases where the risk of reinfection is high [22]. The antibacterial coatings of implants can be classified into passive surface finishing/modifications, active surface finishing/modifications and perioperative antibacterial local carriers or coatings [22]. Silver coatings are one of the most used coatings and we must classify them within active surface finishing/modifications (inorganic) [22]. Most studies agree that silver-coated prostheses reduce the risk of infection without increasing silver-related toxicity [23,24,25]. Scoccianti et al. [23] describes a series of 33 patients (previous infection in 21 patients and high risk for infection because of local or general conditions in 12 patients) operated with megaprostheses with an innovative peripheral silver-added layer of titanium alloy (‘PorAg^®^’) (Hamburg, Germany). Only two patients suffered a recurrence of the infection. Wafa et al. [24] conducted a case-control study comparing 85 patients with Agluna^®^-treated (silver-coated) (Oxfordshire, United Kingdom) tumor implants with 85 control patients with identical, but uncoated, tumor prostheses. There were 50 primary reconstructions, 79 one-stage revisions and 41 two-stage revisions for infection. The overall post-operative infection rate of the silver-coated group was 11.8% compared with 22.4% for the control group. Recently, a meta-analysis [25] with 19 studies was published, noting that silver-coated implants reduce the risk of infection when these are indicated as either a primary indication or a revision. Overall infection rate in primary silver-coated megaprosthesis was 9.2%, compared to 11.2% of non-silver-coated implants and the overall infection rate after revisions was 13.7% in patients with silver-coated megaprosthesis and 29.2% with uncoated megaprosthesis. With these results, silver-coated megaprosthesis would be a good indication when we have to perform one-stage revision arthroplasty with resection of the proximal femur.

A basic limitation of our study is the number of cases; however, the presence of radiological evident diffuse osteomyelitis of the proximal femur in association with chronic PJI is a very uncommon situation.

In conclusion, our experience shows that in chronic PJI associated with diffuse proximal femoral osteomyelitis cases, one-stage exchange arthroplasty with megaprosthesis is a viable option based on the knowledge of the expected functional result and survivorship of this kind of prosthesis. However, there is no evidence that one-stage protocol is superior to two-stage protocol in cases of chronic periprosthetic infection requiring bone resection of the proximal femur.

## Figures and Tables

**Figure 1 tropicalmed-07-00005-f001:**
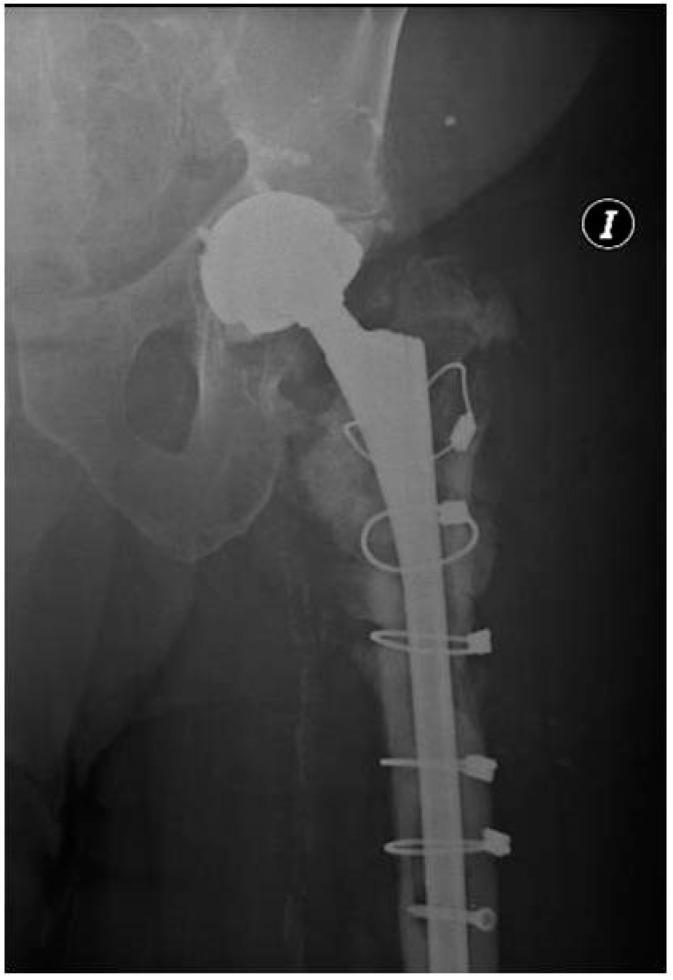
Hip radiography showing proximal femoral diffuse osteomyelitis and signs of non-union of his previous periprosthetic fracture.

**Figure 2 tropicalmed-07-00005-f002:**
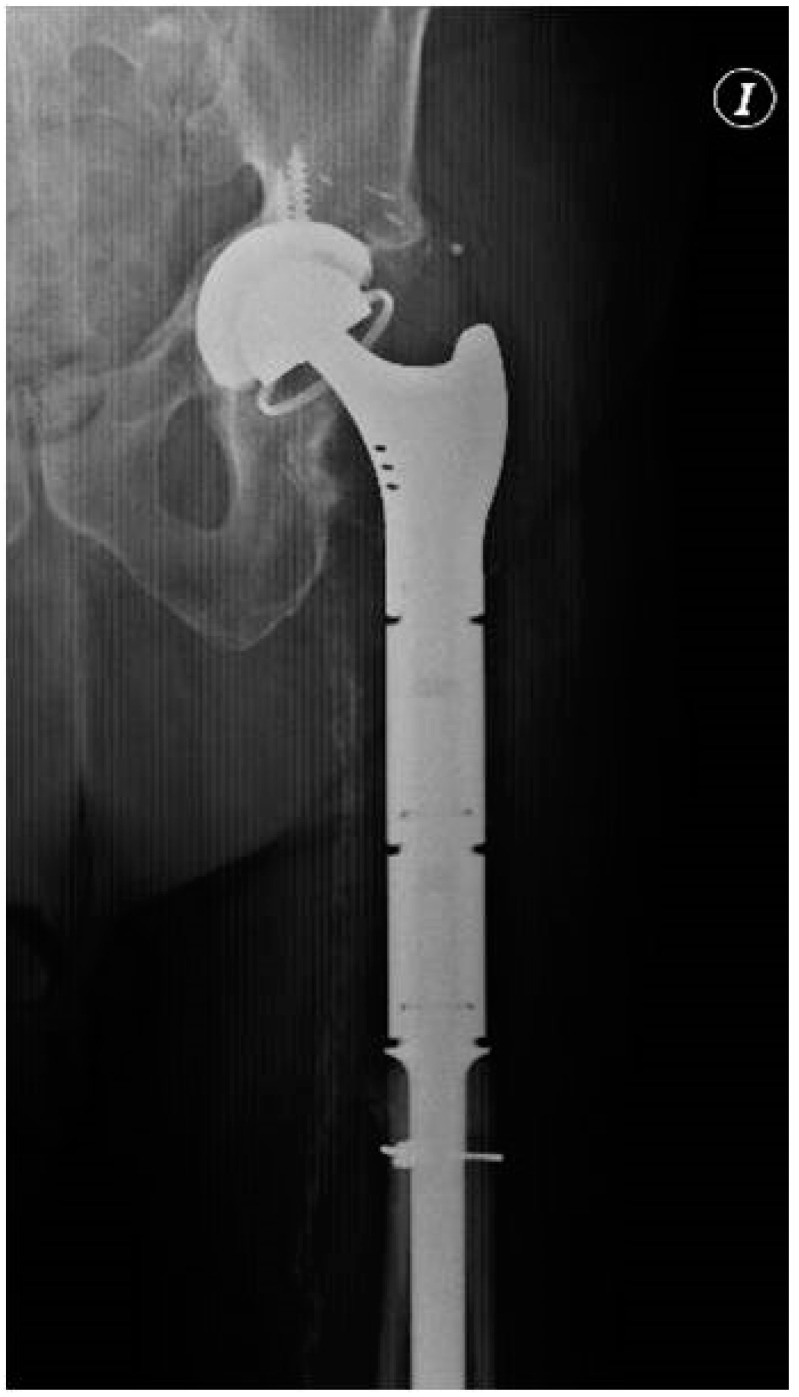
Hip radiography showing the implanted prosthesis.

**Figure 3 tropicalmed-07-00005-f003:**
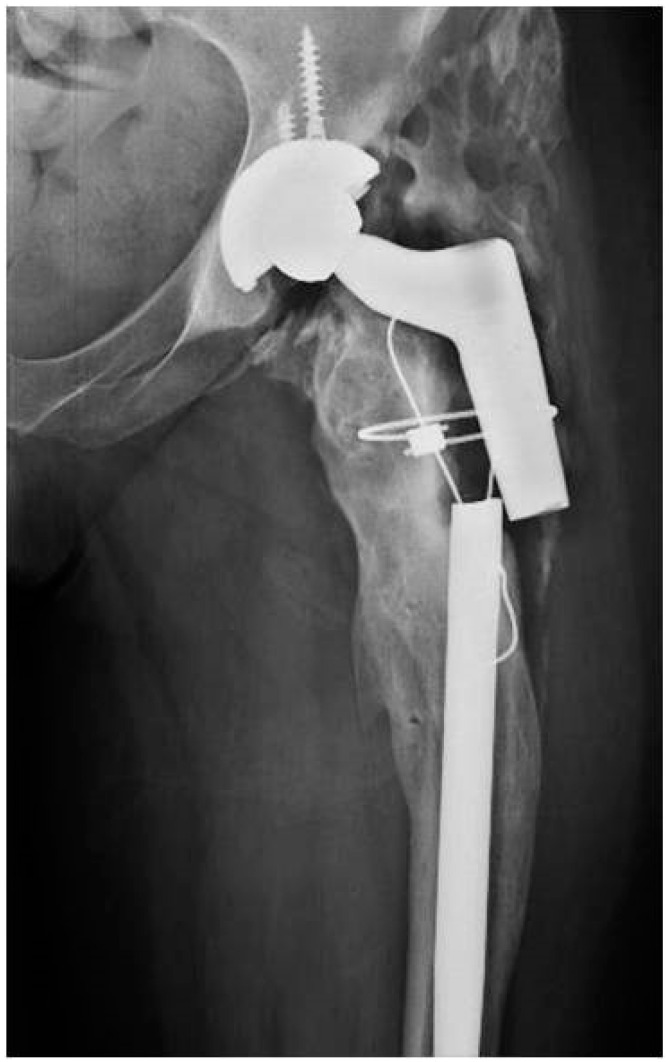
Hip radiography showing diffuse osteomyelitis of the proximal femur and femoral stem rupture.

**Figure 4 tropicalmed-07-00005-f004:**
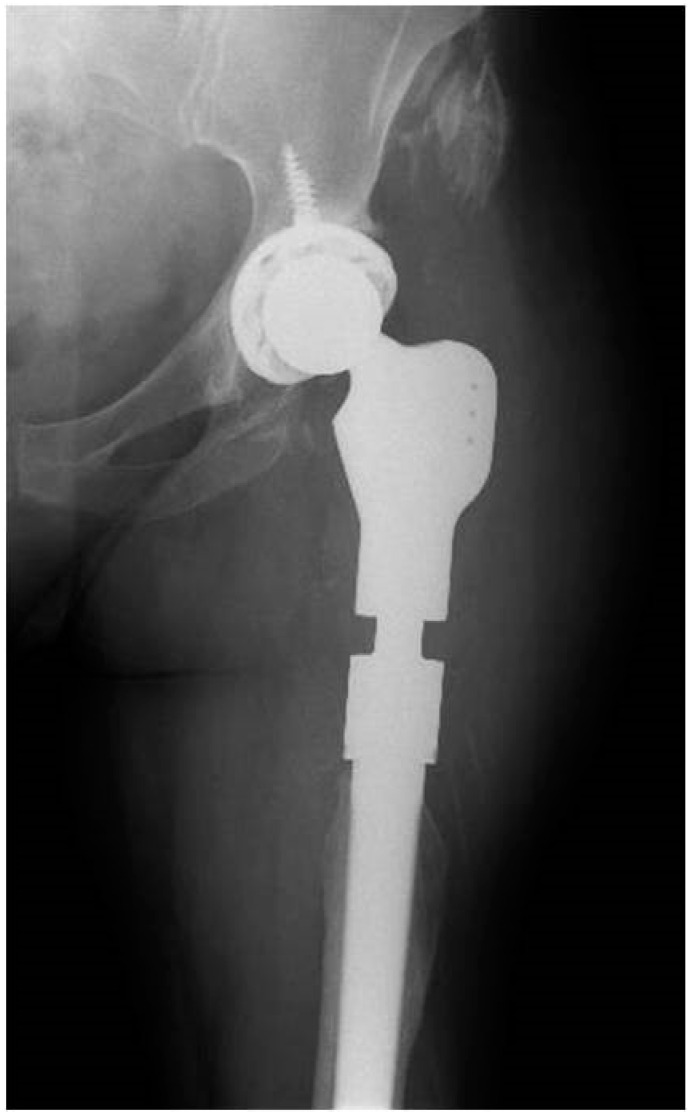
Hip radiology with implanted megaprosthesis with four-years follow-up.

## Data Availability

The datasets used in the current study are available from the corresponding author on reasonable request.

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
