# Peer review of "One-Stage Hip Revision Arthroplasty Using Megaprosthesis in Severe Bone Loss of The Proximal Femur Due to Radiological Diffuse Osteomyelitis"

_tropicalmed, 2021, doi:10.3390/tropicalmed7010005_

Round 1
Reviewer 1 Report
Thank you for giving me the opportunity to review this work. This is a paper about the use of a one-stage procedures by megaprostheses in peri-implant osteomyelitis of the proximal femur.This is a gray area of the literature and the use of this technique appears to be justified even if little used. The two case reports are well presented, complete and have adequate follow-up. The discussion is fairly articulated even if it does not fully cover the subject. In my opinion, the article is ready to publication.
Author Response
Reviewer 1
Thank you for giving me the opportunity to review this work. This is a paper about the use of a one-stage procedures by megaprostheses in peri-implant osteomyelitis of the proximal femur.
This is a gray area of the literature and the use of this technique appears to be justified even if little used. The two case reports are well presented, complete and have adequate follow-up. The discussion is fairly articulated even if it does not fully cover the subject. In my opinion, the article is ready to publication.
Thanks for reviewing the manuscript.
Reviewer 2 Report
Thank you for the opportunity to review this article. The article describes two cases of chronic PJI of the hip treated with one-stage protocol and implantation of proximal femur megaprosthesis. The topic is very interesting and the article well-written. The one-stage protocol in the treatment of PJI offers unquestionable possible advantages, and defining the criteria for selecting patients with PJI who can be treated with one-stage is, in my opinion, a priority in the field of current orthopaedics. In particular, there is little evidence of results in cases of massive bone loss requiring megaprosthesis. In fact, in the two cases described, the choice of one-stage may seem hazardous, as these cases had all the hypothetical characteristics to address the treatment towards the two-stage protocol. Therefore, I would like to ask the authors to clarify some aspects.
1) The use of the one-stage protocol in these two cases, despite the potential risk factors of failure, deserves to be better motivated. Why did the authors decide to use the one-stage protocol?
2) The excellent treatment results, despite the potential risk factors for failure, lead to the conclusion that the 'demolitive' aspect of the PJI revision surgery, aimed at surgically debriding the tissues, plays a key role in improving outcomes. The more material that is removed, the more radical the procedure, the higher the probability of success. In these two cases, the need for extensive resection of the femur with all the potentially bacterial reservoir material may have contributed to the success. The "demolitive" attitude may be able to partially explain the paradox of a very good result in such an unfavourable condition. I would like to see such considerations introduced in the Discussion, in case the Authors agree.
3) In line with the previous comment, I would like to see a very detailed description of the surgical aspects of the cases.
4) The conclusions are too strong and in my opinion should be mitigated. I think we can conclude that the one-stage protocol can be considered even in such complex cases, if it is considered possible to obtain a radical debridement, as there is no strong evidence of a superiority of the two-stage protocol in cases of PJI requiring large bone resection.
Author Response
Reviewer 2
Thank you for the opportunity to review this article. The article describes two cases of chronic PJI of the hip treated with one-stage protocol and implantation of proximal femur megaprosthesis. The topic is very interesting and the article well-written. The one-stage protocol in the treatment of PJI offers unquestionable possible advantages, and defining the criteria for selecting patients with PJI who can be treated with one-stage is, in my opinion, a priority in the field of current orthopaedics. In particular, there is little evidence of results in cases of massive bone loss requiring megaprosthesis. In fact, in the two cases described, the choice of one-stage may seem hazardous, as these cases had all the hypothetical characteristics to address the treatment towards the two-stage protocol. Therefore, I would like to ask the authors to clarify some aspects.
1.- The use of the one-stage protocol in these two cases, despite the potential risk factors of failure, deserves to be better motivated. Why did the authors decide to use the one-stage protocol?
We decided on the one-stage replacement because it was the protocol that existed at the time for the treatment of chronic periprosthetic infection. Our hospital had carried out two-stage until 2013, and only in particular cases was it replaced in one-stage. From 2013 we changed the protocol, and we started to treat ALL the chronic hip infections with one-stage, regardless of germ, fistula, bone defect, ... How we had to perform the one-stage revision arthroplasty, the only way to cure these patients, seeing that they had this radiological osteomyelitis, was to perform a resection of the proximal femur.
2.- The excellent treatment results, despite the potential risk factors for failure, lead to the conclusion that the 'demolitive' aspect of the PJI revision surgery, aimed at surgically debriding the tissues, plays a key role in improving outcomes. The more material that is removed, the more radical the procedure, the higher the probability of success. In these two cases, the need for extensive resection of the femur with all the potentially bacterial reservoir material may have contributed to the success. The "demolitive" attitude may be able to partially explain the paradox of a very good result in such an unfavourable condition. I would like to see such considerations introduced in the Discussion, in case the Authors agree.
The reflection by the reviewer is very interesting, and the article published this november 2021 by the endoklinik group (1) answers many of these questions. We have added a new text in the discussion because in this november 2022 (while we were submitting the article to your journal) the Endoklinik group has published a very interesting article on Resection of the proximal femur during one stage revision for infected hip arthroplasty. An article very similar to ours but with a much higher number of patients. In this study they say that "To date, no study has analyzed potential risk factors and results of resection of the proximal femur as part of a one-stage revision procedure for PJI of the hip." The new text added is as follows:
The Endoklinik group recently published a study on resection of the proximal femur during one stage review for infected hip arthroplasty. It is a case-control study, 57 patients who underwent one-stage revision arthroplasty for PJI of the hip and required resection of the proximal femur and the control group consisted of 57 patients undergoing one-stage revision without bony resection. They conclude that radical resection is associated with higher surgical complications and increased re-revision rates, as patients who required resection of the proximal femur were found to have a higher all-cause re-revision rate (29.8% vs. 10.5%), largely due to reinfection (15.8% vs 0%). However, radical resection may be necessary and required for infection eradication, given the complexity of such septic revision arthroplasty. In these patients it is very important to avoid dislocation of the prosthesis, as in both our cases a constrained liner was implanted, as in this study they conclude that postoperative dislocation were associated with increased risk of subsequent re-revisions with an OR of 281.4. For this reason, Abdelaziz et al found that the use of dual mobility components / constrained liner in the resection group was higher than that of controls (94.7% vs 36.8%). An interesting fact of this study is how they explain that the decision to perform the resection of the proximal femur can be made preoperatively or intraoperatively.
1.- Abdelaziz H, Schröder M, Shum Tien C, Ibrahim K, Gehrke T, Salber J, Citak M. Resection of the proximal femur during one-stage revision for infected hip arthroplasty. Bone Joint J. 2021;103-B(11):1678-1685.
3.- In line with the previous comment, I would like to see a very detailed description of the surgical aspects of the cases.
We've added this text to the discussion:
In both cases, it was decided preoperatively that one-stage revision arthroplasty would be performed with resection of the proximal femur due to radiographic imaging of osteomyelitis. The level of resection was decided intraoperatively based on macroscopic bone findings. We did not use any other system to decide the level of resection, nor the intraoperative bone frozen section (1,2) nor the fluorescent tetracycline bone labeling (3).
1.- Morgenstern M, Athanasou NA, Ferguson JY, Metsemakers MJ, Atkins BL, McNally MA. The value of quantitative histology in the diagnosis of fracture-related infection. Bone Joint J. 2018;100-B(7):966-972.
2.- Bori G, McNally MA, Athanasou N. Histopathology in Periprosthetic Joint Infection: When Will the Morphomolecular Diagnosis Be a Reality? Biomed Res Int. 2018 May 13;2018:1412701.
3.- Muñoz-Mahamud E, Fernández-Valencia JA, Combalia A, Morata L, Soriano A.
Fluorescent tetracycline bone labeling as an intraoperative tool to debride necrotic bone during septic hip revision: a preliminary case series. J. Bone Joint Infect. 2021;6:85-90.
4.- The conclusions are too strong and in my opinion should be mitigated. I think we can conclude that the one-stage protocol can be considered even in such complex cases, if it is considered possible to obtain a radical debridement, as there is no strong evidence of a superiority of the two-stage protocol in cases of PJI requiring large bone resection.
In conclusion we have added the following text:
However, there is no evidence that one-stage protocol is superior to two-stage protocol in cases of chronic periprosthetic infection requiring bone resection of the proximal femur.

Reviewer 3 Report
Good paper describing the results of one-stage exchange arthroplasty using megaprostheses.
I would like the authors to add a paragraph in the discussion on the use of silver-coated megaprostheses in this setting of infection with massive bone defects. It seems to be the most appropriate reconstructive option here.
Line 101: Can you please clarify what was meant by (no antibiotic prophylaxis was given)? At what stage excatly was no prophylactic antibiotic given?
Line 161: delete space in the phrase (Most (24/40) of the patients....).
Author Response
Reviewer 3
Good paper describing the results of one-stage exchange arthroplasty using megaprostheses.
I would like the authors to add a paragraph in the discussion on the use of silver-coated megaprostheses in this setting of infection with massive bone defects. It seems to be the most appropriate reconstructive option here.
1.- We add a paragraph:
From a surgical point of view, we have used conventional megaprosthesis, without any special coating in order to prevent infection, despite the risk that these patients had. There are currently different antibacterial coating of implants on the market that could be useful in those cases where the risk of reinfection is high (1). The antibacterial coating of implants can be classified into passive surface finishing / modifications, active surface finishing / modifications and perioperative antibacterial local carriers or coatings (1). Silver coatings are one of the most used coatings and we must classify them within active surface finishing / modifications (inorganic) (1). Most studies agree that silver-coated prostheses reduce the risk of infection without increasing silver-related toxicity (2,3,4). Scoccianti et al (2) describes a series of 33 patients (previous infection in 21 patients and high risk for infection because of local or general conditions in 12 patients) operated with megaprosthese with an innovative peripheral silver-added layer of titanium alloy (‘PorAg®’). Only two patients suffered a recurrence of the infection. Wafa et al (3) conducted a case-control study comparing 85 patients with Agluna®-treated (silver-coated) tumour implants with 85 control patients with identical, but uncoated, tumour prostheses. There were 50 primary reconstructions, 79 one-stage revisions and 41 two-stage revisions for infection. The overall post-operative infection rate of the silver-coated group was 11.8% compared with 22.4% for the control group. Recently, a meta-analysis (4) with 19 studies was published, noting that silver-coated implants reduce the risk of infection when these are indicated as either a primary indication or a revision. Overall infection rate in primary silver-coated megaprothesis was been 9.2%, compared to 11.2% of non-silver-coated implants and the overall infection rate after revisions was 13.7% in patients with silver-coated megaprosthesis and 29.2% when uncoated megaprostheis. With these results, silver-coated megaprothesis would be a good indication when we have to perform one-stage revision arthroplasty with resection of the proximal femur.
2.- We have added a new text in the discussion because in this november 2022 (while we were submitting the article to your journal) the Endoklinik group has published a very interesting article on Resection of the proximal femur during one stage revision for infected hip arthroplasty. An article very similar to ours but with a much higher number of patients. In this study they say that "To date, no study has analyzed potential risk factors and results of resection of the proximal femur as part of a one-stage revision procedure for PJI of the hip." The new text added is as follows:
The Endoklinik group (0) recently published a study on resection of the proximal femur during one stage review for infected hip arthroplasty. It is a case-control study, 57 patients who underwent one-stage revision arthroplasty for PJI of the hip and required resection of the proximal femur and the control group consisted of 57 patients undergoing one-stage revision without bony resection. They conclude that radical resection is associated with higher surgical complications and increased re-revision rates, as patients who required resection of the proximal femur were found to have a higher all-cause re-revision rate (29.8% vs. 10.5%), largely due to reinfection (15.8% vs 0%). However, radical resection may be necessary and required for infection eradication, given the complexity of such septic revision arthroplasty. In these patients it is very important to avoid dislocation of the prosthesis, as in both our cases a constrained liner was implanted, as in this study they conclude that postoperative dislocation were associated with increased risk of subsequent re-revisions with an OR of 281.4. For this reason, Abdelaziz et al found that the use of dual mobility components / constrained liner in the resection group was higher than that of controls (94.7% vs 36.8%). An interesting fact of this study is how they explain that the decision to perform the resection of the proximal femur can be made preoperatively or intraoperatively.
New references:
0.- Abdelaziz H, Schröder M, Shum Tien C, Ibrahim K, Gehrke T, Salber J, Citak M. Resection of the proximal femur during one-stage revision for infected hip arthroplasty. Bone Joint J. 2021;103-B(11):1678-1685.
1.- Romanò CL, Tsuchiya H, Morelli I, Battaglia AG, Drago L. Antibacterial coating of implants: are we missing something? Bone Joint Res 2019;8:199-206.
2.- Scoccianti G, Frenos F, Beltrami G, Campanacci DA, Capanna R. Levels of silver ions in body fluids and clinical results in silver-coated megaprostheses after tumour, trauma or failed arthroplasty. Injury. 2016;47S:S11-S16.
3.- Wafa H, Grimer RJ, Reddy K, Jeys L, Abudu A, Carter SR, Tillman RM. Retrospective evaluation of the incidence of early periprosthetic infection with silvertreated endoprostheses in high-risk patients. Bone Joint J. 2015;97-B:252–7.
4.- Fiore M, Sambri A, Zucchini R, Giannini C, Donati DM, De Paolis M. Silver‑coated megaprosthesis in prevention and treatment of peri‑prosthetic infections: a systematic review and meta‑analysis about efficacy and toxicity in primary and revision surgery. European Journal of Orthopaedic Surgery & Traumatology. 2021;31:201–220
Line 101: Can you please clarify what was meant by (no antibiotic prophylaxis was given)? At what stage excatly was no prophylactic antibiotic given?
The reviewer is right. This phrase can be confusing and has changed. The patient did not receive antibiotics before the surgery. Prophylaxis began just before surgery, and Meropenem and Linezolid were used. The text has been changed and replaced by:
“therefore, he was not given antibiotic treatment before surgery”
“which was started 15 minutes before surgery”
Line 161: delete space in the phrase (Most (24/40) of the patients....).
The space was deleted.

Round 2
Reviewer 2 Report
I think that the changes have improved overall quality of the article, which is suitable for publication in the present form, in my opinion. Thank you.